# Trained-MPC: A Private Inference by Training-Based Multiparty Computation

Hamidreza Ehteram [1]   Mohammad Ali Maddah-Ali [2]   Mahtab Mirmohseni [3] [1]

## Abstract

How can we perform inference on data using cloud servers without leaking any information to them? The answer lies in *Trained-MPC*, an innovative approach to inference privacy that can be applied to deep learning models. It relies on a cluster of servers, each running a learning model, which are fed with the client data added with **strong** noise. The noise is independent of user data, but **dependent** across the servers. The variance of the noise is set to be large enough to make the information leakage to the servers negligible. The dependency among the noise of the queries allows the parameters of the models running on different servers to be **trained** such that the client can mitigate the contribution of the noises by combining the outputs of the servers, and recover the final result with high accuracy and with a minor computational effort. In other words, in the proposed method, we develop a multiparty computation (MPC) by training for a specific inference task while avoiding the extensive communication overhead that MPC entails. Simulation results demonstrate Trained-MPC resolves the tension between privacy and accuracy while avoiding the computational and communication load needed in cryptography schemes.

## 1 Introduction

With the expansion of machine learning (ML) applications, which deal with high-dimensional datasets and models, it is inevitable to offload heavy computational and storage tasks to cloud servers, particularly for resource-constrained edge devices (e.g., mobile units). The vision of future 6G networks is to enable the edge nodes to send their data to the servers, so the servers can perform the inference and send the results back (Uusitalo et al., 2021). This raises a list of challenges, such as communication overhead, operation cost, etc. One of the major concerns, becoming increasingly important, is maintaining the privacy of the client data such that the level of information leakage to the cloud servers is under control.

In this paper, we focus on *the inference privacy problem*, where a client wishes to employ some server(s) to run an already trained model on his data while preserving the privacy of his data against curiosity of servers. There are various techniques to provide privacy in ML, with the following three major categories:

**(I) Randomization, Perturbation, and Adding Noise:** Applying these techniques to the client data confuses the servers and reduces the level of information leakage, at the cost of sacrificing the accuracy of the results. The information leakage can be measured using concepts such as mutual information (Cover & Thomas, 2012) and differential privacy (Dwork & Roth, 2014). The authors in (Li et al., 2017; Liu et al., 2017a; Wang et al., 2018a; Osia et al., 2020; Mireshghallah et al., 2020; 2021) partition a deep neural network between edge and cloud and offload the computation of some layers to the server. Those papers only consider input privacy and do not guarantee output privacy, i.e., some labels of the input data are exposed to the server. In addition, their performance depends on heavy computation on the client-side, e.g., over 40% of the computation is still performed by the client in (Osia et al., 2018).

**(II) Secure Multiparty Computation (MPC):** This approach exploits the existence of a cluster of servers, some of them non-colluding, to guarantee information-theoretic privacy in some classes of computation tasks like the polynomial functions (Shamir, 1979; Yao, 1982; Ben-Or et al., 1988). This approach can be applied to an ML algorithm (Mohassel & Zhang, 2017; So et al., 2019; Wagh et al., 2019; 2020; Koti et al., 2021). The shortcoming of this solution is that it costs the network a huge communication overhead due to the need for exchanging encrypted data, intermediate results, and coordination messages between the parties. Moreover, some specifications in the setup of this approach (e.g., polynomial approximation and finite field arithmetic)

[1] Department of Electrical Engineering, Sharif University of Technology [2] Department of Electrical and Computer Engineering, University of Minnesota Twin Cities [3] Institute for Communication Systems, University of Surrey. Correspondence to: Mohammad Ali Maddah-Ali <maddah@umn.edu>.

*Proceedings of the $6^{th}$ MLSys Conference Workshop on Resource-Constrained Learning in Wireless Networks*, Miami, FL, USA, 2023. Copyright 2023 by the author(s).

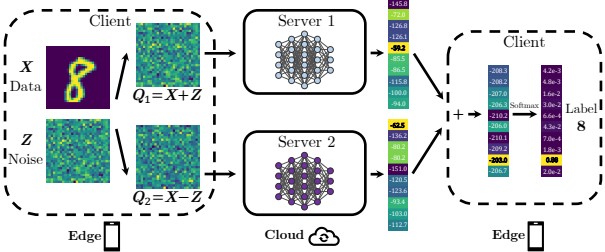

Figure 1: A motivating example of Trained-MPC framework for inference privacy. By injecting the correlated queries and combining the answers with a simple addition, the client obtains the result with high confidence, as confirmed by the softmax vector. Thanks to the very strong noise with a standard deviation of 70 times that of the data, the servers cannot infer anything about the data and its label (preserving both input and output privacy). The high performance of Trained-MPC comes from the fact that the information leakage from each individual query is negligible, while the joint queries inject useful information into the system for classification.

make it challenging to use in deep learning.

**(III) Homomorphic Encryption (HE):** Homomorphic encryption (Gentry & Boneh, 2009) is a cryptography method that can be applied for ML applications (Gilad-Bachrach et al., 2016; Hesamifard et al., 2017; Han et al., 2019). It creates a cryptographically secure framework, between the client and the servers, that allows the untrusted servers to process the encrypted data directly. However, the computational overhead of HE schemes is very high due to the complex mathematical operations and specialized algorithms on encrypted data (with a much larger size than plain data). In addition, it is based on computational hardness assumption and does not guarantee information-theoretic privacy.

**Our contributions:** In this paper, we propose Trained-MPC as an alternative approach to preserve privacy in offloading ML algorithms in a multi-server setup. Our *key idea* is to design the queries of the servers in such a way that they individually do not leak any information to the servers but jointly inject the data into the system so that we can achieve a high level of privacy and accuracy simultaneously. For this purpose, we use *correlated queries* (see Figure 1). These correlated queries are generated by adding strong noises to the client data, where the added noises are independent of the data itself but dependent across the servers. On one hand, we set a large enough noise variance to guarantee negligible information leakage to the servers, ensuring both *input and output privacy*. On the other hand, the dependency among the noise of the queries provides an opportunity for the parameters of the models in different servers to be trained in a way that the client, by combining the outputs of the servers, can mitigate the contribution of the noises. This opportunity allows the system to potentially

have *high accuracy* even when we preserve almost perfect privacy. Note that such an opportunity does not basically exist in the approaches using adding noise and perturbations. Furthermore, Trained-MPC is very *efficient*. In this method, each server runs a regular ML model (e.g., a deep neural network) with no computational overhead. In addition, there is absolutely no communication among the servers. Indeed, the servers may not even be aware of the existence of each other. Also, Trained-MPC achieves high accuracy with very few servers. The experimental results demonstrate that the proposed method significantly outperforms the adding noise approach and ARDEN (Wang et al., 2018a), a framework based on perturbation and adding noise techniques.

In a nutshell, Trained-MPC stems multiparty computation (MPC) by training for a specific inference task, resolving the tension between privacy and accuracy, and avoiding the extensive communication and computational overhead of MPC and HE approaches, respectively.

The rest of the paper is organized as follows. Section 2 formally presents Trained-MPC framework. Section 3 details a design for Trained-MPC in the learning task. In Section 4, the experimental results are discussed.

## 2 GENERAL TRAINED-MPC FRAMEWORK

**Policy:** In Trained-MPC, we want to provide a *private machine-learning-as-a-service*. We consider a system including a client, with limited computational and storage resources, and $N$ servers. The client has individual data and wishes to run an ML algorithm (e.g., deep neural networks) on it with the aid of the servers, while he wishes to keep his data private from the servers. For this purpose, the client sends queries to the servers, and then by combining the received answers, he derives the target (e.g., label in the supervised learning). Here, the client data and its target are both sensitive.

**Threat model:** This paper considers a *semi-honest setup* with $N$ honest-but-curious servers. All the servers follow the protocol, but an arbitrary subset of up to $T < N$ of them may collude (by sharing their queries together) to gain information about the client data. As mentioned earlier, one of the interesting aspects of Trained-MPC framework is that the servers do not need to communicate with each other - in other words, the framework does not impose any communication among the servers on the system. Indeed, the servers may not even be aware of the existence of each other. This aspect makes our setup quite practical. Also, the threat model assumes that any curious server can access unlimited computational resources to extract information about the client data. In other words, we preserve *information-theoretic privacy* in this paper rather than only computational privacy.

**System model:** The system is operated in two phases, the training phase, and then the inference phase.

In the training phase, the dataset $\mathcal{S}_m = \{(\boldsymbol{X}^{(1)}, \boldsymbol{Y}^{(1)}), \ldots, (\boldsymbol{X}^{(m)}, \boldsymbol{Y}^{(m)})\}$ consisting of $m \in \mathbb{N}$ samples is used by the client to train the model, where $(\boldsymbol{X}^{(i)}, \boldsymbol{Y}^{(i)})$ denotes the data sample and its target, for $i = 1, \ldots, m$. In addition, the client generates $m$ independent and identically distributed (i.i.d.) noise samples $\mathcal{Z}_m = \{(\boldsymbol{Z}_1^{(1)}, \ldots, \boldsymbol{Z}_N^{(1)}), \ldots, (\boldsymbol{Z}_1^{(m)}, \ldots, \boldsymbol{Z}_N^{(m)})\}$, where each noise sample $\boldsymbol{Z}^{(i)} = (\boldsymbol{Z}_1^{(i)}, \ldots, \boldsymbol{Z}_N^{(i)})$, with $N$ *correlated* components, is sampled from a joint distribution $\mathbb{P}_{\mathbf{Z}} = \mathbb{P}_{\boldsymbol{Z}_1, \ldots, \boldsymbol{Z}_N}$. The noise components are independent of the dataset $\mathcal{S}_m$.

For $i = 1, \ldots, m$, the client, having access to the dataset $\mathcal{S}_m$ and the noise component set $\mathcal{Z}_m$, uses a preprocessing function $g_{\mathsf{Pre}}$ to generate $N$ queries $\left(Q_1(\boldsymbol{X}^{(i)}, \boldsymbol{Z}_1^{(i)}), \ldots, Q_N(\boldsymbol{X}^{(i)}, \boldsymbol{Z}_N^{(i)})\right) = g_{\mathsf{Pre}}(\boldsymbol{X}^{(i)}, \boldsymbol{Z}^{(i)})$ and sends $Q_j(\boldsymbol{X}^{(i)}, \boldsymbol{Z}_j^{(i)})$ to the $j$-th server, for $j = 1, \ldots, N$. In response, the $j$-th server applies a function (a model) $f_j$ (which will be trained), and generates the answer $\boldsymbol{A}_j^{(i)}$ as $\boldsymbol{A}_j^{(i)} = f_j(Q_j(\boldsymbol{X}^{(i)}, \boldsymbol{Z}_j^{(i)}))$. By combining all the answers from the $N$ servers using a post-processing function $g_{\mathsf{Post}}$, the client estimates the target, $\hat{\boldsymbol{Y}}^{(i)} = g_{\mathsf{Post}}(\boldsymbol{A}_1^{(i)}, \ldots, \boldsymbol{A}_N^{(i)})$, while the information leakage from the set of queries to any arbitrary $T$ servers must be negligible.

In the training phase, the goal is to design or train the set of functions $\mathcal{F} = \{g_{\mathsf{Pre}}, g_{\mathsf{Post}}, f_1, \ldots, f_N\}$ and $\mathbb{P}_{\mathbf{Z}}$ according to the following optimization problem,

$$\min_{\mathcal{F}, \mathbb{P}_{\mathbf{Z}}} \quad \frac{1}{m} \sum_{i=1}^{m} \mathcal{L}\{\hat{\boldsymbol{Y}}^{(i)}, \boldsymbol{Y}^{(i)}\}$$
$$\text{s.t.} \quad \Gamma\left(\boldsymbol{X}^{(i)}; Q_{\mathcal{T}}(\boldsymbol{X}^{(i)}, \boldsymbol{Z}^{(i)})\right) \le \varepsilon, \qquad (1)$$
$$\forall \mathcal{T} \subset [N], |\mathcal{T}| \le T,$$
$$\forall i \in [m],$$

where $Q_{\mathcal{T}}(\boldsymbol{X}^{(i)}, \boldsymbol{Z}^{(i)}) \triangleq \{Q_j(\boldsymbol{X}^{(i)}, \boldsymbol{Z}_j^{(i)}), j \in \mathcal{T}\}$ and $\mathcal{L}\{\hat{\boldsymbol{Y}}^{(i)}, \boldsymbol{Y}^{(i)}\}$ shows the loss function between $\hat{\boldsymbol{Y}}^{(i)}$ and $\boldsymbol{Y}^{(i)}$, for some loss function $\mathcal{L}$. $\Gamma$ denotes the leakage function and measures the privacy leakage, which can be defined according to mutual information (MI) or differential privacy (DP). The constraint guarantees that information leakage through any set of $T$ queries is less than $\varepsilon \in \mathbb{R}_{\ge 0}$. Furthermore, we desire that the computational and storage costs of $g_{\mathsf{Pre}}$ and $g_{\mathsf{Post}}$ are low.

In the inference phase, to deploy this model to estimate the target of a new input $\boldsymbol{X}$, the client chooses $(\boldsymbol{Z}_1, \ldots, \boldsymbol{Z}_N)$, sampled from designed distribution $\mathbb{P}_{\mathbf{Z}}$, independent of all

other variables in the network, and follows the same protocol and uses the designed or trained functions set $\mathcal{F}$.

**Privacy measure:** We measure privacy with both criterions MI and DP. Here, let $g_{\mathsf{Pre}}$ be a mechanism with a random source $\mathbf{Z}$ sampled from the distribution $\mathbb{P}_{\mathbf{Z}} = \mathbb{P}_{\boldsymbol{Z}_1, \ldots, \boldsymbol{Z}_N}$ that takes an independent variable $\boldsymbol{X}$ as input and generates $N$ queries, i.e., $\left(Q_1(\boldsymbol{X}, \boldsymbol{Z}_1), \ldots, Q_N(\boldsymbol{X}, \boldsymbol{Z}_N)\right) = g_{\mathsf{Pre}}(\boldsymbol{X}, \mathbf{Z})$.

**Definition 1** (Mutual information privacy preserving)**.** *Let $\boldsymbol{X}$ be a random variable sampled from an arbitrary distribution $\mathbb{P}_{\boldsymbol{X}}$. The mechanism $g_{\mathsf{Pre}}$ satisfies $\varepsilon$-MI privacy if for all $\mathcal{T} \subset [N]$ of size $T$ it holds that:*

$$I\left(\boldsymbol{X}; Q_{\mathcal{T}}(\boldsymbol{X}, \mathbf{Z})\right) \le \varepsilon.$$

This definition uses Shannon's mutual information (Cover & Thomas, 2012) to measure privacy. This definition states that information obtained about the client data by observing any set of $T$ released queries is negligible. Here, $\varepsilon \in \mathbb{R}_{\ge 0}$ is the privacy parameter; the less $\varepsilon$, the more privacy is preserved. Besides mutual information, differential privacy (Dwork & Roth, 2014) is a well-known privacy metric in the machine-learning context. Although DP is mainly used for revealing information about the training dataset in the trained model privacy problem, we also consider this metric by the following definition.

**Definition 2** (Strict differential privacy preserving)**.** *The mechanism $g_{\mathsf{Pre}}$ satisfies $(\varepsilon, \delta)$-SDP privacy if for all $\mathcal{T} \subset [N]$ of size $T$ the following inequality holds for any two arbitrary inputs $\boldsymbol{X}$ and $\boldsymbol{X}'$ and for any possible output set $\mathcal{O}$ of the corresponding $T$ queries:*

$$\mathbb{P}\left[Q_{\mathcal{T}}(\boldsymbol{X}, \mathbf{Z}) \in \mathcal{O}\right] \le e^{\varepsilon} \mathbb{P}\left[Q_{\mathcal{T}}(\boldsymbol{X}', \mathbf{Z}) \in \mathcal{O}\right] + \delta,$$

*where the probability space is over the random source of the mechanism.*

This definition states that change of the client data in the input has a negligible effect on the distribution of any set of $T$ released queries. Here, $\varepsilon \in \mathbb{R}_{\ge 0}$ and $\delta \in [0, 1]$ are the privacy parameters; the less $(\varepsilon, \delta)$, the more privacy is preserved. Note that since strict differential privacy is defined for any two inputs (which can be different in one pixel or a subset of pixels or completely different), it guarantees conventional differential privacy defined for two adjacent inputs.

## 3 DETAILS OF ONE DESIGN

In this section, we propose one realization of Trained-MPC. The structure of the algorithm has been shown in Figure 2. Here, we explain the different components of the proposed algorithm and then we analyze privacy and accuracy.

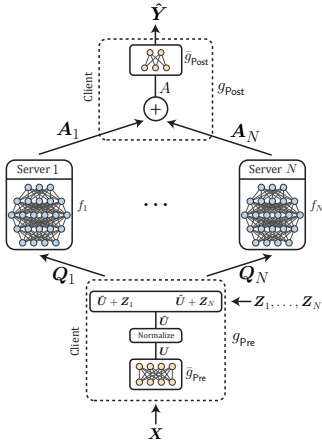

Figure 2: Trained-MPC for classification

## 3.1 Algorithm

**Correlated joint distribution** $\mathbb{P}_{\mathbf{Z}}$**:** The following steps describe how we generate samples of $\mathbf{Z}$. First a matrix $W \in \mathbb{R}^{T \times N}$ is formed, such that (i) any submatrix of size $T \times T$ of W is *full rank*, and (ii) for any submatrix $\Omega$ of size $T \times (T+1)$ of W, the matrix $[\vec{1}, \Omega^\top]$ is *full rank*, here $\vec{1}$ denotes the all-ones vector with length $T+1$. In addition, random matrix $\bar{\mathbf{Z}} = [\bar{\mathbf{Z}}_1, \ldots, \bar{\mathbf{Z}}_T] \in \mathbb{R}^{s \times T}$, independent of $\mathbf{X}$, is formed where each entry is chosen independently and identically from $\mathcal{N}(0, \sigma^2)$. Here, $s$ is the size of each query and $\sigma^2$ is a positive real number, denoting the variance of each entry. Then, let $\mathbf{Z}$ be

$$\mathbf{Z} \triangleq [\mathbf{Z}_1, \ldots, \mathbf{Z}_N] = \bar{\mathbf{Z}} W. \tag{2}$$

As will see, conditions (i) guarantees privacy-preserving, and condition (ii) potentially provide the chance of noise cancellation at the client (accuracy-preserving).

**Preprocessing function** $g_{\mathsf{Pre}}$**:** This function is formed as,

$$\begin{aligned} \mathbf{Q}_j = Q_j(\mathbf{X}, \mathbf{Z}_j) &\triangleq G(\mathbf{X}) + \mathbf{Z}_j \\ &= \mathsf{Normalized}(\bar{g}_{\mathsf{Pre}}(\mathbf{X})) + \mathbf{Z}_j. \end{aligned} \tag{3}$$

Since the client has limited computing resources, $\bar{g}_{\mathsf{Pre}}$ can be a neural network with one layer or even an identity function. $\mathsf{Normalized}(\cdot)$ is defined in Appendix A.

As we will see in Subsection 3.2, a large enough noise variance $\sigma^2$ is sufficient to make the constraint of Optimization (1) be satisfied, independent of the choice of $\bar{g}_{\mathsf{Pre}}$ function.

**Post-processing function** $g_{\mathsf{Post}}$**:** We form $g_{\mathsf{Post}}$ by running a neural network with learnable parameters, denoted by $\bar{g}_{\mathsf{Post}}$, over the sum of the received answers from the servers. Therefore, $\hat{\mathbf{Y}} = \bar{g}_{\mathsf{Post}}(\mathbf{A}) = \bar{g}_{\mathsf{Post}}\left(\sum_{j=1}^N \mathbf{A}_j\right)$. To limit the computational burden on the client, $\bar{g}_{\mathsf{Post}}$ is chosen as at most a single-layer neural notwork with learnable parameters.

**Functions** $f_1$ **to** $f_N$**:** These functions are chosen as some

*multi-layer neural networks* with learnable parameters. The parameters of $f_1$ to $f_N$ will be different.

**Training:** To train the learnable parameters of $\mathcal{F} = \{g_{\mathsf{Pre}}, g_{\mathsf{Post}}, f_1, \ldots, f_N\}$, we use some a particular form of gradient descent optimization algorithms to minimize the loss of Optimization (1). In other words, we train a model, consisting of $N$ separate neural networks $f_1$ to $f_N$ and two networks $\bar{g}_{\mathsf{Pre}}$ and $\bar{g}_{\mathsf{Post}}$. The details of this method are presented in Algorithm 1 (see Appendix B). In this algorithm, the parameters of the model are denoted by $\theta$ and the training batch size is indicated by $b$.

### 3.2 Privacy and Accuracy Analysis

Theorem 1 and 2 respectively show that the proposed method satisfies $\varepsilon$-MI and $(\varepsilon, \delta)$-SDP privacy if we choose the standard deviation $\sigma$ large enough. In the following two theorems, $s$ is the size of each query and

$$p \triangleq \max_{\Omega \in \mathcal{W}} \{\vec{1}^\top (\Omega^\top \Omega)^{-1} \vec{1}\} \in \mathbb{R}_+, \tag{4}$$

where $\mathcal{W}$ denotes the set of all $T \times T$ submatrices of W and $\vec{1}$ denotes the all-ones vector with length $T$. $\Phi$ is the CDF of the standard normal.

**Theorem 1** ($\varepsilon$-MI privacy)**.** *Let* $\left(Q_1(\mathbf{X}, \mathbf{Z}_1), \ldots, Q_N(\mathbf{X}, \mathbf{Z}_N)\right) = g_{\mathsf{Pre}}(\mathbf{X}, \mathbf{Z})$ *be the mechanism as defined in* (2) *and* (3)*. The mechanism* $g_{\mathsf{Pre}}$ *satisfies $\varepsilon$-MI privacy if* $\sigma \geq \frac{\sqrt{ps}}{\sqrt{2 \ln 2}} \frac{1}{\sqrt{\varepsilon}}$*.*

**Theorem 2** ($(\varepsilon, \delta)$-SDP privacy)**.** *Let* $\left(Q_1(\mathbf{X}, \mathbf{Z}_1), \ldots, Q_N(\mathbf{X}, \mathbf{Z}_N)\right) = g_{\mathsf{Pre}}(\mathbf{X}, \mathbf{Z})$ *be the mechanism as defined in* (2) *and* (3)*. The mechanism* $g_{\mathsf{Pre}}$ *satisfies* $(\varepsilon, \delta)$*-SDP privacy if* $\Phi(\frac{\sqrt{ps}}{\sigma} - \frac{\sigma \varepsilon}{2\sqrt{ps}}) - e^\varepsilon \Phi(-\frac{\sqrt{ps}}{\sigma} - \frac{\sigma \varepsilon}{2\sqrt{ps}}) \leq \delta$*.*

Proof of Theorems 1 and 2 and the accuracy analysis can be found in Appendix C. The experiments in Section 4 show that by choosing $\sigma$ large enough, neither the client data nor its true label can be learned by any $T$ servers.

## 4 EXPERIMENTS

This section is dedicated to reporting the performance of the proposed method for the classification task. The implementation details and the network structure are presented in Appendix D.1.

### 4.1 Privacy-Accuracy-Efficiency Trade-Off

We evaluate the performance of Trained-MPC for $N = 2$ and $T = 1$, for three different datasets (MNIST (LeCun et al., 2010), Fashion-MNIST (Xiao et al., 2017), and Cifar-10 (Krizhevsky, 2009)) and for various noise levels, and we report the test accuracy. Here we choose $W_{1 \times 2} = [1, -1]$.

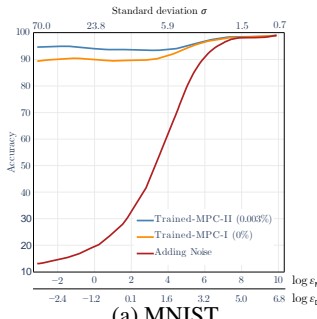
(a) MNIST

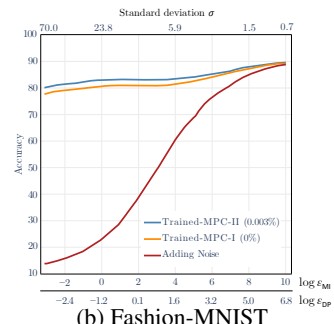
(b) Fashion-MNIST

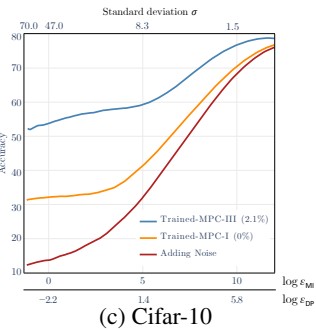
(c) Cifar-10

Figure 3: The Privacy and Accuracy Curves

We also compare the performance of our method with the adding noise approach, where in this case there is only one server and noise is added to the input to protect the privacy of the data. This system is trained to label the noisy data.

Figure 3 demonstrates both the accuracy of the proposed method and the accuracy of the system with one server versus $\sigma$ from 0 to 70, $\log \varepsilon_{\text{MI}}$ for MI privacy, and $\log \varepsilon_{\text{DP}}$ for normalized Strict-DP privacy with $\delta_{\text{SDP}} = 10^{-5}$ (defined in Appendix C.4). For each dataset, we evaluate our method for two models. In the Trained-MPC-I model, the client has no neural networks for $\bar{g}_{\text{Pre}}$ and $\bar{g}_{\text{Post}}$ functions, i.e., the computational load on the client is almost nothing. In Trained-MPC-II and Trained-MPC-III, there is only one layer neural network for $\bar{g}_{\text{Post}}$ or $\bar{g}_{\text{Pre}}$, respectively (the details of the network structure are provided in Experiment 3 of Appendix D). Also, the computational cost of the client relative to the computational cost of the entire network is written next to each model (computational complexity is calculated by the number of required products.).

This figure shows that, unlike the systems with one server, Trained-MPC achieves good accuracy for various datasets, even for a high noise level. For example, in Figure 3a, the client with a minor post-processing in Trained-MPC-II achieves 95% accuracy while the privacy leakage is less than $\varepsilon_{\text{MI}} = 0.115$ and $\varepsilon_{\text{DP}} = 0.121$, thanks to the intense noise with $\sigma = 70$. In contrast, with a single server and adding noise with the same variance, we can reach 13% accuracy. In general, although the accuracy of Trained-MPC decreases with increasing the noise variance, it still converges to a reasonable value; on the contrary, the adding noise approach has no gain in perfect privacy preserving. In summary, Trained-MPC provides a superior trade-off between privacy-accuracy-efficiency compared to the conventional approaches.

## 4.2 Comparison

We compare Trained-MPC with the adding noise approach and ARDEN (Wang et al., 2018a), which is a framework based on perturbation and adding noise techniques. ARDEN partitions a neural network across edge and cloud and trans-

| | ARDEN (Wang et al.) | Adding Noise | Trained-MPC-II (Ours) |
|---|---|---|---|
| **Accuracy** Classification Rate of Client | 91.3% | 91.1% | **95.1%** |
| **Computation** Percentage on Client | 36.7% | 0% | **0.003%** |
| **Output Privacy** Misclassification Rate of Server | 8.7% | 8.9% | **86.5%** |
| **Input Privacy** Reconstruction Loss to Data Power | 25.9% | 16.1% | **46.4%** |
| **Query** Transmitted From Client to Server | | | |
| **Attack** Reconstructed Query by Server | Attacked | Attacked | Defended |

Figure 4: Comparison. The approaches based on perturbation and adding noise have no effective way to mitigate added confusion and suffer from the level of accuracy and privacy. The noise mitigation opportunity in Trained-MPC enables the client to achieve high accuracy in perfect privacy with a minor computation.

mits the noisy representation of data to the server. It trains the system on a mixture of the noisy representation and its clean and perturbed version. As its model parameters are learnable, it becomes more robust to noise than approaches like (Mireshghallah et al., 2020; 2021), which do not retrain the model.

In ARDEN implementation, we use its suggested hyperparameters (of (Wang et al., 2018a)) and allocate the first two layers of the network to the client-side. Figure 4 compares the three methods on the MNIST dataset. Moreover, it visualizes the query and its reconstructed version by the server for an identical input sample. We leverage the autoencoder implemented in (Ni, 2018) for the reconstruction and use mean squared error (MSE) as the loss function.

The figure shows ARDEN, at the cost of computational burden on the client, improves input privacy slightly compared to the adding noise approach; however, Trained-MPC outperforms ARDEN by 4% higher accuracy, 10X more privacy, and 4 orders of magnitude fewer computations on the client. The low costs on the client-side and the guarantee of good accuracy make Trained-MPC framework suitable for practical use cases in mobile devices and the internet of things (IoT). We report more experiment results in Appendix D.2.

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

# Appendix

The appendix is organized as follows: Appendix A provides the notations. Appendix B presents the proposed algorithm. Appendix C is dedicated to the proofs. Appendix D provides the experimental details and simulation results. Appendix E reviews the relead works.

## A   NOTATIONS

Capital italic bold letter $\boldsymbol{X}$ denotes a random vector. Capital non-italic bold letter $\mathbf{X}$ denotes a random matrix. Capital non-italic non-bold letter X denotes a deterministic matrix. $I(\boldsymbol{X}; \boldsymbol{Y})$ indicates the mutual information between the two random vectors $\boldsymbol{X}$ and $\boldsymbol{Y}$. For a function $g$, the computational cost (e.g., the number of multiplications) and storage cost (e.g., the number of parameters) are denoted as $C_c(g)$ and $C_s(g)$, respectively. $\mathsf{Normalized}(X)$, for $X = [x_1, \ldots, x_n]^\top \in \mathbb{R}^n$, is defined as $\frac{X - \mu}{\sigma}$, where $\mu = \frac{1}{n} \sum_{i=1}^{n} x_i$ and $\sigma^2 = \frac{1}{n} \sum_{i=1}^{n} (x_i - \mu)^2$. $\log x$ is calculated in base 2 (i.e., $\log_2 x$). For $T \in \mathbb{N}$, $[T] = \{1, \ldots, T\}$. $\mathcal{N}(\mu, \sigma^2)$ denotes Gaussian distribution with mean $\mu$ and variance $\sigma^2$. $\mathrm{W}[:, \{t_1, \ldots, t_T\}]$ denotes a submatrix of W, consisting of the columns $t_1, \ldots, t_T$ of matrix W, respectively. $\boldsymbol{X} \perp\!\!\!\perp \boldsymbol{Y}$ indicates the two random vectors $\boldsymbol{X}$ and $\boldsymbol{Y}$ are independent.

## B   ALGORITHM

---

**Algorithm 1** Trained-MPC: Inference privacy with $N$ servers, up to $T$ colluding

---

**input:** $\mathcal{S}_m, \sigma, \mathrm{W}, \bar{g}_{\mathsf{Pre}}(\cdot; \theta_{\bar{g}_{\mathsf{Pre}}}), \bar{g}_{\mathsf{Post}}(\cdot; \theta_{\bar{g}_{\mathsf{Post}}}), f_j(\cdot; \theta_{f_j}), b$
**output:** $\mathbb{P}_{\mathbf{Z}}$ and updated $\bar{g}_{\mathsf{Pre}}(\cdot; \theta_{\bar{g}_{\mathsf{Pre}}}), \bar{g}_{\mathsf{Post}}(\cdot; \theta_{\bar{g}_{\mathsf{Post}}}), f_j(\cdot; \theta_{f_j})$

$i \leftarrow 1, \ldots, b$
$s \leftarrow$ the output size of $\bar{g}_{\mathsf{Pre}}(\cdot; \theta_{\bar{g}_{\mathsf{Pre}}})$
**function** $\mathbb{P}_{\mathbf{Z}}$
    Draw $s \times T$ *i.i.d.* noise samples from $\mathcal{N}(0, \sigma^2)$
    Shape the noise samples to $s \times T$ matrix $\bar{\mathbf{Z}}$
    Compute the noise components: $[\boldsymbol{Z}_1, \ldots, \boldsymbol{Z}_N] \leftarrow \bar{\mathbf{Z}}\mathrm{W}$
    **return** $(\boldsymbol{Z}_1, \ldots, \boldsymbol{Z}_N)$
**end**
**for** *the number of training iterations* **do**
    **Forward path:**
    Draw $b$ minibatch samples from $\mathcal{S}_m$: $\{(\boldsymbol{X}^{(1)}, \boldsymbol{Y}^{(1)}), \ldots, (\boldsymbol{X}^{(b)}, \boldsymbol{Y}^{(b)})\}$
    Draw $b$ *i.i.d.* noise samples from $\mathbb{P}_{\mathbf{Z}}$: $\{(\boldsymbol{Z}_1^{(1)}, \ldots, \boldsymbol{Z}_N^{(1)}), \ldots, (\boldsymbol{Z}_1^{(b)}, \ldots, \boldsymbol{Z}_N^{(b)})\}$
    Compute the client features: $\boldsymbol{U}^{(i)} \leftarrow \bar{g}_{\mathsf{Pre}}(\boldsymbol{X}^{(i)}; \theta_{\bar{g}_{\mathsf{Pre}}})$
    Normalize the client features: $\hat{\boldsymbol{U}}^{(i)} \leftarrow \mathsf{Normalized}(\boldsymbol{U}^{(i)})$
    **for** $j = 1, \ldots, N$ **do**
        Compute the query of the $j$-th server: $\boldsymbol{Q}_j^{(i)} \leftarrow \hat{\boldsymbol{U}}^{(i)} + \boldsymbol{Z}_j^{(i)}$
        Compute the answer of the $j$-th server: $\boldsymbol{A}_j^{(i)} \leftarrow f_j(\boldsymbol{Q}_j^{(i)}; \theta_{f_j})$
    **end**
    Compute the sum of the answers: $\boldsymbol{A}^{(i)} \leftarrow \sum_{j=1}^{N} \boldsymbol{A}_j^{(i)}$
    Compute the client predicted labels: $\hat{\boldsymbol{Y}}^{(i)} \leftarrow \bar{g}_{\mathsf{Post}}(\boldsymbol{A}^{(i)}; \theta_{\bar{g}_{\mathsf{Post}}})$
    Compute the loss: $L(\theta_{\bar{g}_{\mathsf{Pre}}}, \theta_{\bar{g}_{\mathsf{Post}}}, \theta_{f_1}, \ldots, \theta_{f_N}) \leftarrow \frac{1}{b} \sum_i \mathcal{L}\{\hat{\boldsymbol{Y}}^{(i)}, \boldsymbol{Y}^{(i)}\}$
    **Backward path:**
    The backpropagation (BP) algorithm:
    Update $\theta_{\bar{g}_{\mathsf{Post}}}$
    **for** $j = 1, \ldots, N$ **do**
        Compute the BP of the client to the $j$-th server
        Update $\theta_{f_j}$
        Compute the BP of the $j$-th server to the client
    **end**
    Update $\theta_{\bar{g}_{\mathsf{Pre}}}$
**end**

---

As it is clear, the servers do not need to communicate with each other in the forward path. The following proposition states it also holds for the backward.

**Proposition B.1.** *The servers do not need to communicate with each other in the backward path of Algorithm 1.*

*Proof.* We calculate the gradient of the loss function with respect to the model parameters for a given data $\boldsymbol{X}$. Let $l = \mathcal{L}\{\hat{\boldsymbol{Y}}, \boldsymbol{Y}\}$ be the loss for the data $\boldsymbol{X}$. Using denominator-layout notation, we have:

$$\frac{\partial l}{\partial \theta_{\bar{g}_{\text{Post}}}} = \overbrace{\frac{\partial \hat{\boldsymbol{Y}}}{\partial \theta_{\bar{g}_{\text{Post}}}} \frac{\partial l}{\partial \hat{\boldsymbol{Y}}}}^{\text{computed by client}},$$

$$\frac{\partial l}{\partial \theta_{f_j}} = \frac{\partial \boldsymbol{A}}{\partial \theta_{f_j}} \frac{\partial l}{\partial \boldsymbol{A}} = \frac{\partial (\boldsymbol{A}_1 + \ldots + \boldsymbol{A}_N)}{\partial \theta_{f_j}} \frac{\partial l}{\partial \boldsymbol{A}} \overset{(*)}{=} \overbrace{\frac{\partial \boldsymbol{A}_j}{\partial \theta_{f_j}}}^{\text{computed by } j\text{-th server}} \overbrace{\frac{\partial l}{\partial \boldsymbol{A}_j}}^{\text{BP of client to } j\text{-th server}},$$

$$\frac{\partial l}{\partial \theta_{\bar{g}_{\text{Pre}}}} = \frac{\partial \boldsymbol{A}}{\partial \theta_{\bar{g}_{\text{Pre}}}} \frac{\partial l}{\partial \boldsymbol{A}} = \sum_j \frac{\partial \boldsymbol{A}_j}{\partial \theta_{\bar{g}_{\text{Pre}}}} \frac{\partial l}{\partial \boldsymbol{A}_j} = \sum_j \frac{\partial \boldsymbol{Q}_j}{\partial \theta_{\bar{g}_{\text{Pre}}}} \frac{\partial \boldsymbol{A}_j}{\partial \boldsymbol{Q}_j} \frac{\partial l}{\partial \boldsymbol{A}_j} = \sum_j \overbrace{\frac{\partial \boldsymbol{Q}_j}{\partial \theta_{\bar{g}_{\text{Pre}}}}}^{\text{computed by client}} \overbrace{\frac{\partial l}{\partial \boldsymbol{Q}_j}}^{\text{BP of } j\text{-th server to client}},$$

where (*) follows from $\frac{\partial l}{\partial \boldsymbol{A}_j} = \frac{\partial \boldsymbol{A}}{\partial \boldsymbol{A}_j} \frac{\partial l}{\partial \boldsymbol{A}} = \text{I} \frac{\partial l}{\partial \boldsymbol{A}} = \frac{\partial l}{\partial \boldsymbol{A}}$ and BP indicates the backpropagation. Clearly, the framework does not impose any communication among the servers on the system. □

## C  PROOF OF PRIVACY AND ACCURACY PRESERVING

### C.1  Proof of Theorem 1

**Theorem C.1** ($\varepsilon$-MI privacy). *Let* $\big(Q_1(\boldsymbol{X}, \boldsymbol{Z}_1), \ldots, Q_N(\boldsymbol{X}, \boldsymbol{Z}_N)\big) = g_{\text{Pre}}(\boldsymbol{X}, \mathbf{Z})$ *be the mechanism as defined in* (2) *and* (3). *The mechanism* $g_{\text{Pre}}$ *satisfies* $\varepsilon$-*MI privacy if*

$$\sigma \geq \frac{\sqrt{ps}}{\sqrt{2 \ln 2}} \frac{1}{\sqrt{\varepsilon}}. \tag{5}$$

*Proof.* In this theorem, we want to show that for all $\mathcal{T} \subset [N]$ of size $T$, we have

$$I(\boldsymbol{X}; \{Q_j(\boldsymbol{X}, \boldsymbol{Z}_j), j \in \mathcal{T}\}) \leq \varepsilon.$$

Let $\text{K} = \mathbb{E}[G(\boldsymbol{X})G(\boldsymbol{X})^T]$ denote the covariance matrix of $G(\boldsymbol{X})$. Since $G(\boldsymbol{X})$ is Normalized, then $\text{tr}(\text{K}) = s$. In addition, consider the set $\mathcal{T} = \{t_1, \ldots, t_T\}$, where $\mathcal{T} \subset [N]$ and $|\mathcal{T}| = T$, also let $\Omega_{\mathcal{T}} = \text{W}[:, \mathcal{T}]$, and $\mathbf{Q}_{\mathcal{T}} = [Q_{t_1}(\boldsymbol{X}, \boldsymbol{Z}_{t_1}), \ldots, Q_{t_T}(\boldsymbol{X}, \boldsymbol{Z}_{t_T})]$. Then, we have

$$\mathbf{Q}_{\mathcal{T}} = [G(\boldsymbol{X}), \bar{\mathbf{Z}}][\vec{1}, \Omega_{\mathcal{T}}^{\top}]^{\top}, \tag{6}$$

where $\bar{\mathbf{Z}}$ is defined in (3). In addition, we define

$$[\omega_1, \ldots, \omega_T] = \vec{1}^{\top} \Omega_{\mathcal{T}}^{-1}. \tag{7}$$

Thus we have,

$$
I(\boldsymbol{X}; \{Q_j(\boldsymbol{X}, \boldsymbol{Z}_j), j \in \mathcal{T}\}) = I(\boldsymbol{X}; \mathbf{Q}_{\mathcal{T}})
$$

$$
\overset{(a)}{=} I(\boldsymbol{X}; \mathbf{Q}_{\mathcal{T}}\Omega_{\mathcal{T}}^{-1}) = h(\mathbf{Q}_{\mathcal{T}}\Omega_{\mathcal{T}}^{-1}) - h(\mathbf{Q}_{\mathcal{T}}\Omega_{\mathcal{T}}^{-1}|\boldsymbol{X})
$$

$$
\overset{(b)}{=} h(\omega_1 G(\boldsymbol{X}) + \bar{\boldsymbol{Z}}_1, \ldots, \omega_T G(\boldsymbol{X}) + \bar{\boldsymbol{Z}}_T) - h(\bar{\mathbf{Z}})
$$

$$
\overset{(c)}{\leq} \sum_{t=1}^{T} \Big( h(\omega_t G(\boldsymbol{X}) + \bar{\boldsymbol{Z}}_t) - h(\bar{\boldsymbol{Z}}_t) \Big)
$$

$$
\overset{(d)}{\leq} \sum_{t=1}^{T} \frac{1}{2} \Big( \log \big( (2\pi e)^s \det(\omega_t^2 \mathrm{K} + \sigma^2 \mathrm{I}_s) \big) - \log(2\pi e \sigma^2)^s \Big)
$$

$$
\overset{(e)}{\leq} \frac{1}{2} \sum_{t=1}^{T} \Big( \log \big( (\frac{1}{\sigma^2})^s (\frac{\mathrm{tr}(\omega_t^2 \mathrm{K} + \sigma^2 \mathrm{I}_s)}{s})^s \big) \Big)
$$

$$
\overset{(f)}{=} \frac{s}{2} \sum_{t=1}^{T} \log(\frac{\omega_t^2 + \sigma^2}{\sigma^2}) \overset{(g)}{\leq} \frac{s}{2\ln 2} \sum_{t=1}^{T} \frac{\omega_t^2}{\sigma^2} \overset{(h)}{\leq} \varepsilon,
$$

where (a) follows since $\Omega_{\mathcal{T}}$ is a full rank matrix; (b) follows from (6), (7) and the fact that $\bar{\mathbf{Z}}$ is independent of $\boldsymbol{X}$; (c) follows from inequality $h(\boldsymbol{A}, \boldsymbol{B}) \leq h(\boldsymbol{A}) + h(\boldsymbol{B})$ for any two random vectors $\boldsymbol{A}$ and $\boldsymbol{B}$, and the fact that the set of random vectors $\{\bar{\boldsymbol{Z}}_1, \ldots, \bar{\boldsymbol{Z}}_T\}$ is mutually independent; (d) follows because $\bar{\mathbf{Z}}$ is independent of $\boldsymbol{X}$ and jointly Gaussian distribution maximizes the entropy of a random vector with a known covariance matrix (Cover & Thomas, 2012); (e) follows from the fact that by considering a symmetric and positive semi-definite matrix $\mathrm{H} = \omega_t^2 \mathrm{K} + \sigma^2 \mathrm{I}_s$ with eigenvalues $\lambda_k$, we have $\det(\mathrm{H}) = \prod_{k=1}^{s} \lambda_k$ and $\mathrm{tr}(\mathrm{H}) = \sum_{k=1}^{s} \lambda_k$, and therefore we obtain $\det(\mathrm{H}) \leq (\frac{\mathrm{tr}(\mathrm{H})}{s})^s$ using the inequality of arithmetic and geometric means; (f) follows since $\mathrm{tr}(\mathrm{K}) = s$; (g) follows due to $\ln(x+1) \leq x$; and (h) follows from (5), (4), (7), and by substituting $\sum_{t=1}^{T} \omega_t^2 = \vec{1}^{\top} (\Omega_{\mathcal{T}}^{\top} \Omega_{\mathcal{T}})^{-1} \vec{1} \leq p$. $\qquad\square$

## C.2  Proof of Theorem 2

**Theorem C.2** (($\varepsilon,\delta$)-SDP privacy). *Let* $\big( Q_1(\boldsymbol{X}, \boldsymbol{Z}_1), \ldots, Q_N(\boldsymbol{X}, \boldsymbol{Z}_N) \big) = g_{\text{Pre}}(\boldsymbol{X}, \mathbf{Z})$ *be the mechanism as defined in* (2) *and* (3). *The mechanism* $g_{\text{Pre}}$ *satisfies* ($\varepsilon,\delta$)-SDP *privacy if*

$$
\Phi(\frac{\sqrt{ps}}{\sigma} - \frac{\sigma\varepsilon}{2\sqrt{ps}}) - e^{\varepsilon} \Phi(-\frac{\sqrt{ps}}{\sigma} - \frac{\sigma\varepsilon}{2\sqrt{ps}}) \leq \delta. \tag{8}
$$

*For* $\delta < \frac{1}{2}$, *Inequality* (8) *yields a simpler bound, but not as tight as before:*

$$
\sigma \geq \sqrt{ps} \big( \frac{-2\Phi^{-1}(\delta)}{\varepsilon} + \frac{1}{-\Phi^{-1}(\delta)} \big). \tag{9}
$$

*Here* $\Phi^{-1}$ *denotes the inverse function of* $\Phi$.

*Proof.* The following steps proof the theorem:

*Step 1. Mechanism and its distribution:*

Consider the set $\mathcal{T} = \{t_1, \ldots, t_T\}$, where $\mathcal{T} \subset [N]$ and $|\mathcal{T}| = T$, and also let $\Omega_{\mathcal{T}} = \mathrm{W}[:, \mathcal{T}]$. All data revealed to the colluding servers is the set $\{Q_j(\boldsymbol{X}, \boldsymbol{Z}_j), j \in \mathcal{T}\}$, where $Q_j(\boldsymbol{X}, \boldsymbol{Z}_j) = G(\boldsymbol{X}) + \boldsymbol{Z}_j$ and $\mathbf{Z}_{\mathcal{T}} \overset{\triangle}{=} [\boldsymbol{Z}_{t_1}, \ldots, \boldsymbol{Z}_{t_T}] = \bar{\mathbf{Z}}\Omega_{\mathcal{T}}$. Indeed, for all set $\mathcal{T}$, we want to evaluate privacy of the mechanism $Q_{\mathcal{T}} : \mathbb{R}^s \to \mathbb{R}^{Ts}$ where $Q_{\mathcal{T}}(\boldsymbol{X}) \overset{\triangle}{=} [G(\boldsymbol{X})^{\top} + \boldsymbol{Z}_{t_1}^{\top}, \ldots, G(\boldsymbol{X})^{\top} + \boldsymbol{Z}_{t_T}^{\top}]^{\top}$. To structure the algebra equations, we use the Kronecker product. The mechanism is rewrited

to $Q_{\mathcal{T}}(\boldsymbol{X}) = \vec{1} \otimes G(\boldsymbol{X}) + \mathsf{vec}(\mathbf{Z}_{\mathcal{T}})$ that has the distribution $\mathcal{N}(\mu_{\boldsymbol{X}}, \Sigma)$. Here,

$$\mu_{\boldsymbol{X}} \overset{(a)}{=} \vec{1} \otimes G(\boldsymbol{X}),$$

$$\begin{aligned}
\Sigma &= \mathbb{E}[\mathsf{vec}(\mathbf{Z}_{\mathcal{T}})\mathsf{vec}(\mathbf{Z}_{\mathcal{T}})^{\top}] \\
&\overset{(b)}{=} \mathbb{E}[(\Omega_{\mathcal{T}}^{\top} \otimes \mathrm{I}_s)\mathsf{vec}(\bar{\mathbf{Z}})\mathsf{vec}(\bar{\mathbf{Z}})^{\top}(\Omega_{\mathcal{T}}^{\top} \otimes \mathrm{I}_s)^{\top}] \\
&\overset{(c)}{=} \sigma^2(\Omega_{\mathcal{T}}^{\top}\Omega_{\mathcal{T}} \otimes \mathrm{I}_s),
\end{aligned}$$

where $\mathsf{vec}(\mathbf{Z}_{\mathcal{T}})$ is a vector obtained by stacking $\mathbf{Z}_{\mathcal{T}}$'s columns; (a) follows since $\mathbf{Z}_{\mathcal{T}}$ is zero-mean; (b) is due to the fact that $\mathsf{vec}(\mathbf{Z}_{\mathcal{T}}) = (\Omega_{\mathcal{T}}^{\top} \otimes \mathrm{I}_s)\mathsf{vec}(\bar{\mathbf{Z}})$ (following from $\mathbf{Z}_{\mathcal{T}} = \bar{\mathbf{Z}}\Omega_{\mathcal{T}}$ with some simple algebra); and (c) follows from $\mathbb{E}[\mathsf{vec}(\bar{\mathbf{Z}})\mathsf{vec}(\bar{\mathbf{Z}})^{\top}] = \sigma^2\mathrm{I}_{Ts}$. As $\Omega_{\mathcal{T}}$ is full rank, the distribution $\mathcal{N}(\mu_{\boldsymbol{X}}, \Sigma)$ is non-degenerate and has density. Note that unlike MI which deals with both the distributions $\mathbb{P}_{\boldsymbol{X}}$ and $\mathbb{P}_{\mathbf{Z}}$, in DP the probability space is over the random source of the mechanism, i.e., the distribution $\mathbb{P}_{\mathbf{Z}}$.

*Step 2. Privacy loss:*

To evaluate privacy of the mechanism $Q_{\mathcal{T}}$, we leverage a necessary and sufficient condition for differential privacy (Balle & Wang, 2018). It states that a mechanism $\mathcal{M}$ satisfies

$$\forall \mathcal{O} : \mathbb{P}[\mathcal{M}(\boldsymbol{X}) \in \mathcal{O}] \le e^{\varepsilon}\mathbb{P}[\mathcal{M}(\boldsymbol{X}') \in \mathcal{O}] + \delta$$

iff we have

$$\mathbb{P}[L_{\mathcal{M},\boldsymbol{X},\boldsymbol{X}'} \ge \varepsilon] - e^{\varepsilon}\mathbb{P}[L_{\mathcal{M},\boldsymbol{X}',\boldsymbol{X}} \le -\varepsilon] \le \delta.$$

$L_{\mathcal{M},\boldsymbol{X},\boldsymbol{X}'}$ is the privacy loss random variable defined as $L_{\mathcal{M},\boldsymbol{X},\boldsymbol{X}'} = \ln\frac{f_{\mathcal{M}(\boldsymbol{X})}(\boldsymbol{Y})}{f_{\mathcal{M}(\boldsymbol{X}')}(\boldsymbol{Y})}$ where the random variable $\boldsymbol{Y}$ follows the distribution of $\mathcal{M}(\boldsymbol{X})$ and $f_{\mathcal{M}(\boldsymbol{X})}(\cdot)$ denotes the probability density function (PDF) of $\mathcal{M}(\boldsymbol{X})$. In the following, we want to calculate the distribution of $L_{Q_{\mathcal{T}},\boldsymbol{X},\boldsymbol{X}'}$ for two arbitrary $\boldsymbol{X}$ and $\boldsymbol{X}'$ and $Q_{\mathcal{T}}(\boldsymbol{X}) \sim \mathcal{N}(\mu_{\boldsymbol{X}}, \Sigma)$.

Consider $\boldsymbol{Y}$ sampled from $\mathcal{N}(\mu_{\boldsymbol{X}}, \Sigma)$. We have:

$$\begin{aligned}
L_{Q_{\mathcal{T}},\boldsymbol{X},\boldsymbol{X}'} &= \ln\frac{f_{Q_{\mathcal{T}}(\boldsymbol{X})}(\boldsymbol{Y})}{f_{Q_{\mathcal{T}}(\boldsymbol{X}')}(\boldsymbol{Y})} \\
&= \ln\frac{\frac{1}{\sqrt{(2\pi)^{Ts}\det(\Sigma)}}\exp\left(-\frac{1}{2}(\boldsymbol{Y}-\mu_{\boldsymbol{X}})^{\top}\Sigma^{-1}(\boldsymbol{Y}-\mu_{\boldsymbol{X}})\right)}{\frac{1}{\sqrt{(2\pi)^{Ts}\det(\Sigma)}}\exp\left(-\frac{1}{2}(\boldsymbol{Y}-\mu_{\boldsymbol{X}'})^{\top}\Sigma^{-1}(\boldsymbol{Y}-\mu_{\boldsymbol{X}'})\right)} \\
&= -\frac{1}{2}\left[(\boldsymbol{Y}-\mu_{\boldsymbol{X}})^{\top}\Sigma^{-1}(\boldsymbol{Y}-\mu_{\boldsymbol{X}}) - (\boldsymbol{Y}-\mu_{\boldsymbol{X}'})^{\top}\Sigma^{-1}(\boldsymbol{Y}-\mu_{\boldsymbol{X}'})\right] \\
&= \frac{1}{2}(\mu_{\boldsymbol{X}}-\mu_{\boldsymbol{X}'})^{\top}\Sigma^{-1}(\mu_{\boldsymbol{X}}-\mu_{\boldsymbol{X}'}) + (\boldsymbol{Y}-\mu_{\boldsymbol{X}})^{\top}\Sigma^{-1}(\mu_{\boldsymbol{X}}-\mu_{\boldsymbol{X}'}).
\end{aligned}$$

Since $\boldsymbol{Y} \sim \mathcal{N}(\mu_{\boldsymbol{X}}, \Sigma)$, $L_{Q_{\mathcal{T}},\boldsymbol{X},\boldsymbol{X}'}$ has the distribution $\mathcal{N}(\eta, 2\eta)$ where $\eta = \frac{1}{2}(\mu_{\boldsymbol{X}}-\mu_{\boldsymbol{X}'})^{\top}\Sigma^{-1}(\mu_{\boldsymbol{X}}-\mu_{\boldsymbol{X}'})$. By substituting $\mu_{\boldsymbol{X}} = \vec{1} \otimes G(\boldsymbol{X})$ and $\Sigma = \sigma^2(\Omega_{\mathcal{T}}^{\top}\Omega_{\mathcal{T}} \otimes I_s)$, we have

$$\eta = \frac{p_{\mathcal{T}}\Delta_{\boldsymbol{X},\boldsymbol{X}'}^2}{2\sigma^2},$$

where $p_{\mathcal{T}} \overset{\triangle}{=} \vec{1}^{\top}(\Omega_{\mathcal{T}}^{\top}\Omega_{\mathcal{T}})^{-1}\vec{1}$ and $\Delta_{\boldsymbol{X},\boldsymbol{X}'} \overset{\triangle}{=} \|G(\boldsymbol{X}) - G(\boldsymbol{X}')\|_2$. In a similar way, $L_{Q_{\mathcal{T}},\boldsymbol{X}',\boldsymbol{X}}$ has the same distribution.

*Step 3. Computing $\varepsilon$ and $\delta$:*

According to the previous step, the mechanism $g_{\mathsf{Pre}}$ is $(\varepsilon, \delta)$-DP iff for all set $\mathcal{T}$ and for any $\boldsymbol{X}$ and $\boldsymbol{X}'$ it holds:

$$\mathbb{P}[\mathcal{N}(\eta, 2\eta) \ge \varepsilon] - e^{\varepsilon}\mathbb{P}[\mathcal{N}(\eta, 2\eta) \le -\varepsilon] \le \delta.$$

Since the left-hand side of the inequality is monotonically increasing function of $\eta$ (see (Balle & Wang, 2018)) , we substitute $\eta$ with the upper bound $\eta = \frac{p_{\mathcal{T}} \Delta^2_{\boldsymbol{X}, \boldsymbol{X}'}}{2\sigma^2} \leq \frac{4ps}{2\sigma^2}$. This bound follows from $p = \max_{\mathcal{T}} p_{\mathcal{T}} = \max_{\Omega \in \mathcal{W}} \{\vec{1}^\top (\Omega^\top \Omega)^{-1} \vec{1}\}$ (see (4)) and $\Delta^2_{\boldsymbol{X}, \boldsymbol{X}'} = \|G(\boldsymbol{X}) - G(\boldsymbol{X}')\|^2_2 \leq 4s$ ($G(\boldsymbol{X})$ is Normalized and $\|G(\boldsymbol{X})\|_2 = \sqrt{s}$ for all $\boldsymbol{X}$). Also, we have $\mathbb{P}[\mathcal{N}(\eta, 2\eta) \geq \varepsilon] = \Phi(\frac{\eta - \varepsilon}{\sqrt{2\eta}})$ and $\mathbb{P}[\mathcal{N}(\eta, 2\eta) \leq -\varepsilon] = \Phi(\frac{-\eta - \varepsilon}{\sqrt{2\eta}})$. Thus, the mechanism $g_{\mathsf{Pre}}$ is $(\varepsilon, \delta)$-DP if

$$\Phi\left(\frac{\sqrt{ps}}{\sigma} - \frac{\sigma\varepsilon}{2\sqrt{ps}}\right) - e^\varepsilon \Phi\left(-\frac{\sqrt{ps}}{\sigma} - \frac{\sigma\varepsilon}{2\sqrt{ps}}\right) \leq \delta,$$

which is obtained from substituting the upper bound of $\eta$ for all set $\mathcal{T}$ in the necessary and sufficient condition.

For $\delta < \frac{1}{2}$, we show that if $\sigma \geq \sqrt{ps}\left(\frac{-2\Phi^{-1}(\delta)}{\varepsilon} + \frac{1}{-\Phi^{-1}(\delta)}\right)$, then the condition given above is satisfied:

$$
\begin{aligned}
\Phi\left(\frac{\sqrt{ps}}{\sigma} - \frac{\sigma\varepsilon}{2\sqrt{ps}}\right) - e^\varepsilon \Phi\left(-\frac{\sqrt{ps}}{\sigma} - \frac{\sigma\varepsilon}{2\sqrt{ps}}\right) &\overset{(a)}{\leq} \Phi\left(\frac{\sqrt{ps}}{\sigma} - \frac{\sigma\varepsilon}{2\sqrt{ps}}\right) \\
&\overset{(b)}{\leq} \Phi\left(\frac{1}{\frac{-2\Phi^{-1}(\delta)}{\varepsilon} + \frac{1}{-\Phi^{-1}(\delta)}} + \Phi^{-1}(\delta) - \frac{\varepsilon}{-2\Phi^{-1}(\delta)}\right) \\
&\overset{(c)}{\leq} \Phi\left(\frac{1}{\frac{-2\Phi^{-1}(\delta)}{\varepsilon}} + \Phi^{-1}(\delta) - \frac{\varepsilon}{-2\Phi^{-1}(\delta)}\right) = \delta,
\end{aligned}
$$

where (a) follows from $\Phi(\cdot) \geq 0$; (b) follows from the fact that $\Phi\left(\frac{\sqrt{ps}}{\sigma} - \frac{\sigma\varepsilon}{2\sqrt{ps}}\right)$ is monotonically decreasing in $\sigma \geq 0$, and we can substitute $\sigma$ with the lower bound in (9); and (c) follows because $\Phi(\cdot)$ is monotonically increasing, and $-\Phi^{-1}(\delta) > 0$ for $\delta < \frac{1}{2}$. $\qquad\square$

## C.3 Proof of Accuracy Preserving

The following theorem shows that the proposed algorithm provides the potential of noise cancellation from every $T + 1$ queries for the client to achieve high accuracy.

**Theorem C.3** (Accuracy Preserving). *Let* $\left(Q_1(\boldsymbol{X}, \boldsymbol{Z}_1), \ldots, Q_N(\boldsymbol{X}, \boldsymbol{Z}_N)\right) = g_{\mathsf{Pre}}(\boldsymbol{X}, \mathbf{Z})$ *be the mechanism as defined in* (2) *and* (3). *For all* $\mathcal{T}' \subset [N]$ *of size* $T + 1$, *there exists a non-constant function* $f$ *such that:*

$$f\left(g_{\mathsf{Pre}_{\mathcal{T}'}}(\boldsymbol{X}, \mathbf{Z})\right) \perp\!\!\!\perp \mathbf{Z}.$$

*Proof.* In this theorem, we want to show that for all $\mathcal{T}' = \{t_1, \ldots, t_{T+1}\}$, there exists a non-constant function $f$ such that:

$$f(Q_{t_1}(\boldsymbol{X}, \boldsymbol{Z}_{t_1}), \ldots, Q_{t_{T+1}}(\boldsymbol{X}, \boldsymbol{Z}_{t_{T+1}})) \perp\!\!\!\perp \mathbf{Z}.$$

According to the definition of matrix W in (3), for all set $\mathcal{T}' = \{t_1, \ldots, t_{T+1}\}$, the matrix $[\vec{1}, \Omega^\top_{\mathcal{T}'}]$ is full rank, where $\Omega_{\mathcal{T}'} = \mathrm{W}[:, \mathcal{T}']$ and $\vec{1}$ is the all-ones vector with length $T + 1$. We claim that

$$f(Q_{t_1}(\boldsymbol{X}, \boldsymbol{Z}_{t_1}), \ldots, Q_{t_{T+1}}(\boldsymbol{X}, \boldsymbol{Z}_{t_{T+1}})) \overset{\triangle}{=} \mathbf{Q}_{\mathcal{T}'}([\vec{1}, \Omega^\top_{\mathcal{T}'}]^\top)^{-1} \vec{e}_1$$

satisfies the problem. Here, $\mathbf{Q}_{\mathcal{T}'} = [Q_{t_1}(\boldsymbol{X}, \boldsymbol{Z}_{t_1}), \ldots, Q_{t_{T+1}}(\boldsymbol{X}, \boldsymbol{Z}_{t_{T+1}})]$ and $\vec{e}_1 = [1, 0, \ldots, 0]^\top$ is the standard basis with length $T + 1$.

Proof of the claim: According to (2) and (3), we have $\mathbf{Q}_{\mathcal{T}'} = [G(\boldsymbol{X}), \bar{\mathbf{Z}}][\vec{1}, \Omega^\top_{\mathcal{T}'}]^\top$. Therefore,

$$
\begin{aligned}
f(Q_{t_1}(\boldsymbol{X}, \boldsymbol{Z}_{t_1}), \ldots, Q_{t_{T+1}}(\boldsymbol{X}, \boldsymbol{Z}_{t_{T+1}})) &= \mathbf{Q}_{\mathcal{T}'}([\vec{1}, \Omega^\top_{\mathcal{T}'}]^\top)^{-1} \vec{e}_1 \\
&= [G(\boldsymbol{X}), \bar{\mathbf{Z}}][\vec{1}, \Omega^\top_{\mathcal{T}'}]^\top ([\vec{1}, \Omega^\top_{\mathcal{T}'}]^\top)^{-1} \vec{e}_1 \\
&= G(\boldsymbol{X}),
\end{aligned}
$$

where $G(\boldsymbol{X})$ is independent of $\mathbf{Z}$. Thus, the system has the chance of noise cancellation. $\qquad\square$

Table 1: Privacy Parameters.

| Dataset | Standard Deviation | Privacy Parameter | |
|---|---|---|---|
| | | $\varepsilon_{\text{MI}}$ | $\varepsilon_{\text{DP}}$ |
| MNIST, Fashion-MNIST | $\sigma = 70$ | 0.1 | 0.1 |
| $s = 28 \times 28$ | $\sigma = 50$ | 0.2 | 0.2 |
| $p = 1$ | $\sigma = 30$ | 0.6 | 0.3 |
| Cifar-10 | $\sigma = 70$ | 0.5 | 0.1 |
| $s = 3 \times 32 \times 32$ | $\sigma = 50$ | 0.9 | 0.2 |
| $p = 1$ | $\sigma = 30$ | 2.5 | 0.4 |

## C.4  Privacy Parameters

Table 1 reports $\varepsilon_{\text{MI}}$ for mutual information privacy and $\varepsilon_{\text{DP}} \triangleq \frac{\varepsilon_{\text{SDP}}}{\sqrt{s}}$ for normalized strict differential privacy with $\delta_{\text{SDP}} = 10^{-5}$ at different values of the standard deviation $\sigma$. Here, we choose $s$ equal to the size of the raw data and $p = 1$ (for $W_{1 \times 2} = [1, -1]$).

**Remark:** As said before, strict differential privacy guarantees conventional differential privacy. A question may arise - why do we define DP for any two inputs instead of two adjacent inputs (e.g., different in only one feature or pixel)? The point is that unlike a usual dataset, which has independent instances, the elements of an image are correlated (see the notion of group privacy for datasets whose instances are correlated (Dwork & Roth, 2014)). That is, changing a concept/object in a data/image changes a large number of pixels. Thus, it makes sense to define DP for any two inputs when we do not have a bound on the number of pixels associated with a concept. The notation of Strict-DP is approximately $\sqrt{s}$ times stronger than conventional DP (see the impact of the data size $s$ in inequalities (8) and (9)).

Here we illustrate the difference between DP and Strict-DP in a simple example. Consider an image $X = [x_1, \ldots, x_s]^\top \in \mathbb{R}^s$ consisting of $s$ pixels, each in the range $[0, 1]$. The Gaussian Mechanism with parameter $\sigma$ adds noise $\mathcal{N}(0, \sigma^2)$ to each of the pixels. Using Classical Gaussian Mechanism (Dwork & Roth, 2014), the output is $(\varepsilon, \delta)$-DP for $\varepsilon, \delta \in (0, 1)$ if $\sigma \geq \frac{\Delta}{\varepsilon}\sqrt{2 \ln \frac{1.25}{\delta}}$, where $\Delta$ is defined as $\max_{\text{adjacent} X, X'} \|X - X'\|_2$. As it is clear, for two adjacent images (i.e., different in only one pixel), we have $\Delta = 1$. Now consider Strict-DP, which is defined for any two images instead of two adjacent images. In this case, $\Delta$ becomes $\max_{\text{any} X, X'} \|X - X'\|_2 = \|[1, \ldots, 1]^\top\|_2 = \sqrt{s}$. Therefore, $\varepsilon_{\text{SDP}}$ is $\sqrt{s}$ times $\varepsilon_{\text{DP}}$ at the same level of $\sigma$.

## D  EVALUATION

This section is dedicated to reporting simulation results. In Subsection D.1, the implementation details of the proposed method is presented. In Subsections D.2, we have conducted experiments to answer the following questions:

1. How is the performance of Trained-MPC in perfect privacy? (Experiment 1 in Part D.2.1).

2. How does noise affect the output? (Experiment 2 in Part D.2.2)

3. How is the accuracy of Trained-MPC compared to the adding noise approach? (Experiment 3 in Part D.2.3)

4. How much do pre- and post-processing impact accuracy? (Experiment 4 in Part D.2.4)

### D.1  The Implementation Details

**Datasets:** We evaluate the proposed algorithm for the classification task on MNIST (LeCun et al., 2010), Fashion-MNIST (Xiao et al., 2017), and Cifar-10 (Krizhevsky, 2009) datasets by using their standard training sets and testing sets. The only used preprocessings on images are Random Crop and Random Horizontal Flip on Cifar-10 training dataset and padding MNIST and Fashion-MNIST images on all sides with zeros of length 2 to fit in a network with input size $32 \times 32$.

**Setup:** We employ Convolutional Neural Networks (CNNs) in $\bar{g}_{\text{Pre}}$, $f_j$, and $\bar{g}_{\text{Post}}$. We initialize the network parameters by Kaiming initialization (He et al., 2015). For each value of the standard deviation of the noise, we continue the learning

Table 2: Network structure. Normalized$(\cdot)$ function, normalizes its input as defined in the notation A. Conv2d parameters represent the number of the input channels, the number of the output channels, the kernel size, the stride, and the padding of a 2D convolutional layer, respectively. FC parameters represent the number of the input neurons and the number of the output neurons of a fully connected layer. BatchNorm2d and BatchNorm1d parameters represent the number of the input channels and the number of the input neurons respectively for batch normalization layer.

| | N(**Iden.** → **Iden.**) | N(**Iden.** → $n_o$) | N($n_i$ → **Iden.**) | N($n_i$ → $n_o$) |
|---|---|---|---|---|
| $\bar{g}_{Pre}$ | Identity function | Identity function | Conv2d $(c_i,n_i,(5,5),3,0)$ BatchNorm2d $(n_i)$ ReLU | Conv2d $(c_i,n_i,(5,5),3,0)$ BatchNorm2d $(n_i)$ ReLU |
| $f_j$ | Normalized $(\cdot)$ Conv2d $(c_i,64,(5,5),3,0)$ BatchNorm2d (64) ReLU Conv2d (64,128,(3,3),1,0) BatchNorm2d (128) ReLU Flatten FC (128*8*8,1024) BatchNorm1d (1024) ReLU FC (1024,10) | Normalized $(\cdot)$ Conv2d $(c_i,64,(5,5),3,0)$ BatchNorm2d (64) ReLU Conv2d (64,128,(3,3),1,0) BatchNorm2d (128) ReLU Flatten FC (128*8*8,1024) BatchNorm1d (1024) ReLU FC (1024,$n_o$) | Normalized $(\cdot)$ Conv2d $(n_i,128,(3,3),1,0)$ BatchNorm2d (128) ReLU Flatten FC (128*8*8,1024) BatchNorm1d (1024) ReLU FC (1024,10) | Normalized $(\cdot)$ Conv2d $(n_i,128,(3,3),1,0)$ BatchNorm2d (128) ReLU Flatten FC (128*8*8,1024) BatchNorm1d (1024) ReLU FC (1024,$n_o$) |
| $\bar{g}_{Post}$ | Identity function | BatchNorm1d $(n_o)$ ReLU FC $(n_o,10)$ | Identity function | BatchNorm1d $(n_o)$ ReLU FC $(n_o,10)$ |

process for 265 epochs. We also use Adam optimizer (Kingma & Ba, 2014) and decrease the learning rate from $10^{-3}$ to $2 \times 10^{-5}$ exponentially during the training. We set the training batch size equal to 128. The models are implemented using PyTorch (Paszke et al., 2019). For $N = 2$ and $T = 1$, we choose

$$W_{1\times 2} = \begin{bmatrix} 1 & -1 \end{bmatrix}.$$

For $N = 3$ and $T = 2$, we choose

$$W_{2\times 3} = \begin{bmatrix} 0 & \sqrt{\frac{3}{4}} & -\sqrt{\frac{3}{4}} \\ 1 & -\frac{1}{2} & -\frac{1}{2} \end{bmatrix}.$$

For $N = 4$ and $T = 3$, we choose

$$W_{3\times 4} = \begin{bmatrix} 0 & \sqrt{\frac{8}{9}} & -\sqrt{\frac{2}{9}} & -\sqrt{\frac{2}{9}} \\ 0 & 0 & \sqrt{\frac{2}{3}} & -\sqrt{\frac{2}{3}} \\ 1 & -\frac{1}{3} & -\frac{1}{3} & -\frac{1}{3} \end{bmatrix}.$$

**Network structure:** $f_j$ is a neural network with several convolutional layers and two fully connected layers with the Rectified Linear Unit (ReLU) activation function (see Table 2 for details). To limit the computational cost of the pre- and post-processings at the client, we use at most one convolutional layer in $\bar{g}_{Pre}$ and at most one fully connected layer in $\bar{g}_{Post}$. In particular, $\bar{g}_{Pre}$ is either an identity function, denoted by Iden., or a convolutional layer with $n_i \in \mathbb{N}$ output channels. In addition, $\bar{g}_{Post}$ can be a fully connected layer, with a vector of length $n_o \in \mathbb{N}$ as the input, and generating 10 outputs, representing 10 different classes. The input vector of length $n_o$ is formed by adding the $N$ vectors of length $n_o$, received the from servers. In addition, we also consider a very simple case for $\bar{g}_{Post}$, where $n_o = 10$ and at the client side, we simply add up the vectors of length $n_o$, received from the servers. In other words, in this case $\bar{g}_{Post}$ is equal to the identity function. We represent the structure of $\bar{g}_{Pre}$ and $\bar{g}_{Post}$ by N($n_i → n_o$), where the number of the output channels at $\bar{g}_{Pre}$ is equal to $n_i$ (with the exception that $n_i = $ Iden. means $\bar{g}_{Pre}$ is the identity function, i.e., $\bar{g}_{Pre}(X) = X$), and the number of the output neurons at $f_j$ is equal to $n_o$ (with the exception that $n_o = $ Iden. means $\bar{g}_{Post}(A) = A$). In Table 2, $c_i$ denotes the number of the input image channels. Since the variance of the input queries can be high (see Equation (3)) we use Normalized$(.)$ function at the first stage of $f_j$. We use the cross-entropy loss function between $Y$ and softmax$(\hat{Y})$ for $\mathcal{L}\{\hat{Y}, Y\}$.

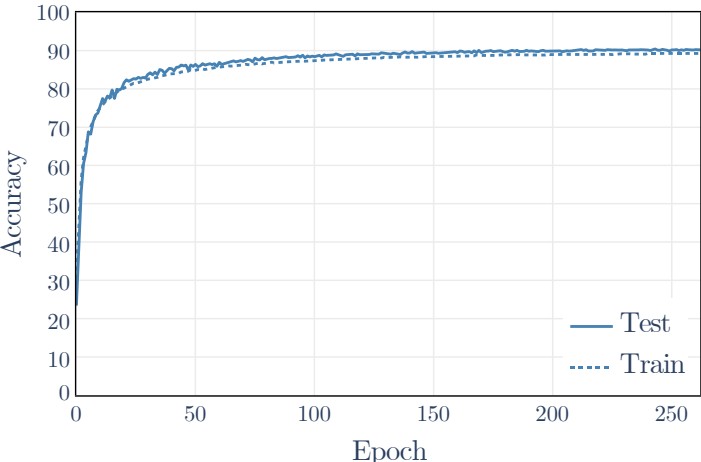

Figure 5: Results on the accuracy versus the epoch number, for noise level $\sigma = 70$, $N = 2$, $T = 1$, and N(Iden. $\rightarrow$ Iden.) model on MNIST dataset in Experiment 1.

## D.2 More Experiments

### D.2.1 Experiment 1: Accuracy Results in Perfect Privacy

Here, we are particularly interested in the case of perfect privacy. For this target, we choose $\sigma = 70$, representing a very strong noise. We consider a vary simple system for the client with $N = 2$ servers and $T = 1$. In addition, we use identity mapping for both $\bar{g}_{\text{Pre}}$ and $\bar{g}_{\text{Post}}$. This means the number of the output neurons at the servers is $10$ and we simply add the received answers of the servers to find the label of the client data. This case is denoted by N(Iden. $\rightarrow$ Iden.). Figure 5 reports the test and train accuracy for the MNIST dataset versus the training epoch number. This figure shows that the system gradually learns how to mitigate the contribution of the correlated noises by combining the outputs of the servers and achieves high accuracy. It shows that using Trained-MPC, the client achieves $90\%$ accuracy while the privacy leakage is less than $\varepsilon_{\text{MI}} = 0.12$, thanks to the strong noise with $\sigma = 70$. Note that it is done without any computational load on the client.

### D.2.2 Experiment 2: Effect of Noise on the Output

To answer the question of how noise affects the output, here we plot the noise distribution in the output. In Figure 6, we use a test sample from MNIST dataset with label "6" as the input. We want to visualize how each server is confused about the correct label (here "6"), with an incorrect one (say "9"). Each plot in the second row of Figure 6 is a 2D-plot histogram, representing the joint distribution of two neurons of the output of server one, i.e., $A_1[6]$ and $A_1[9]$ ($A_1[6]$ on the x-axis and $A_1[9]$ on the y-axis). We have this figure for different values of $\sigma$. If the point $(A_1[6], A_1[9])$ is above the line $y = x$, i.e., $A_1[9] > A_1[6]$, it means that server one incorrectly prefers label "9" to label "6". In the first row in Figure 6, we have the same plots for $A[9]$ versus $A[6]$, where $A = A_1 + A_2$. As we can see, for large noise (i.e., $\sigma = 70$), server one chooses "6" or "9" almost equiprobably, while the client almost always chooses the label correctly. This shows that, in our proposed method, simultaneous training of the two networks $f_1$ and $f_2$ on the correlated queries allows the system to be trained such that the distribution of the noise at the output of the system does not confuse the client about the correct label (see the first row in Figure 6).

### D.2.3 Experiment 3: Correlated vs. Independent Noise

In this part, we express the details of the experiment described in Subsection 4.1. In that experiment, Trained-MPC-I, Trained-MPC-II, and Trained-MPC-III indicate the network structure of N(Iden. $\rightarrow$ Iden.), N(Iden. $\rightarrow$ 32), and N(32 $\rightarrow$ Iden.), respectively. Also, here we plot the accuracy of the proposed method for $N = 2$ and $T = 1$ versus the case with one server in Figures 7. This comparison emphasizes the fact that having more than one server with correlated noise would allow the system to mitigate the negative effects of the noise, even when the noise is strong; however, with one server, even with training, the system cannot eliminate the noise. Thus, the ability of noise mitigation in Trained-MPC follows not only from training in the presence of noise but also from the fact that we inject correlated noises into the servers and have the

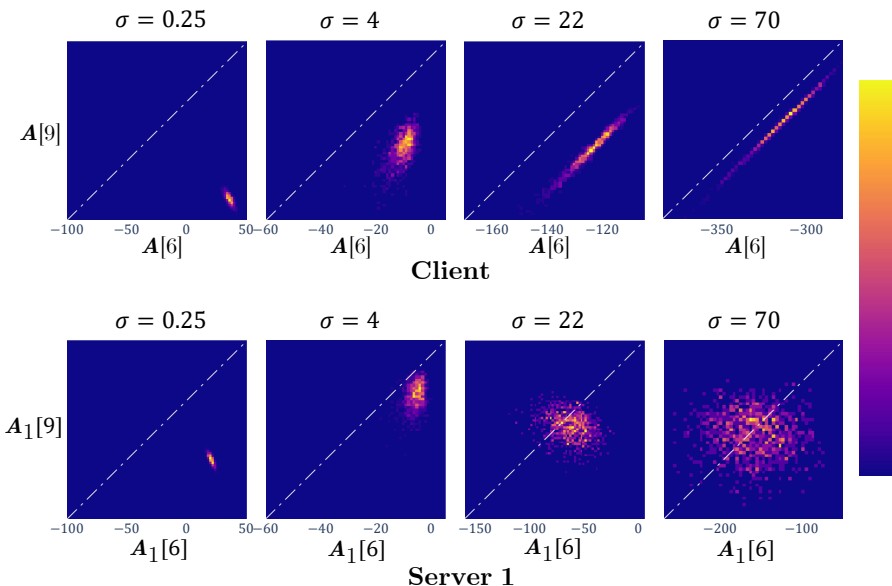

Figure 6: The joint distribution of $(A[6], A[9])$ in the **first row** and $(A_1[6], A_1[9])$ in the **second row** for a sample with label "6" for various the standard deviation of the noise (i.e., $\sigma$) in Expriment 2.

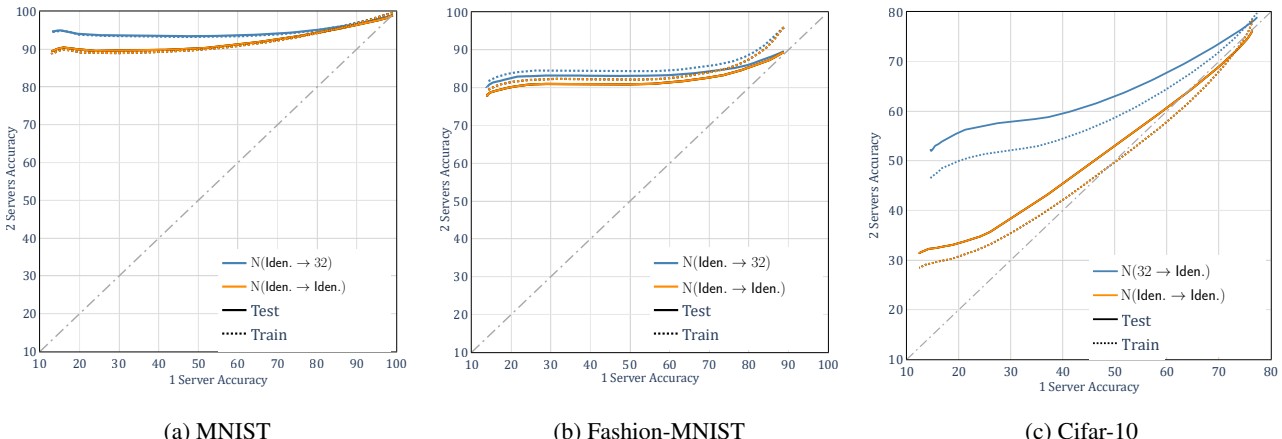

Figure 7: 2 Servers with correlated noise (Trained-MPC) vs. 1 Server (Adding Noise). The results of Experiment 3.

opportunity for noise mitigation.

### D.2.4 Experiment 4: Accuracy vs. Efficiency

In this experiment, we discuss client costs in terms of computation, storage, and communication. In Table 3, we evaluate the proposed algorithm for various models and values $(N, T)$. We report the test accuracy for three datasets and the computational and storage costs of the client relative to the computational and storage costs of one of the servers. In this table, $C_{\mathsf{c}}(\cdot)$ and $C_{\mathsf{s}}(\cdot)$ denote the number of products (representing computational complexity) and the number of the parameters (representing storage complexity) in a model, respectively. In addition, we report the size of each query, denoted by $s$, relative to the size of the client data (the image size).

According to the table, pre- and post-processing results in improved accuracy. Preprocessing plays a role in enhancing accuracy by enabling a higher capacity to project raw data onto a representation that is more conducive to classification. On the other hand, post-processing allows clients to perform additional processing on the combined answers.

Table 3: Test accuracy in the classification task, for various models and different tuple $(N, T)$. The test accuracy for MNIST, Fashion-MNIST, and Cifar-10 is evaluated for $\log \varepsilon_{\text{MI}} = 0$, 0, and 1.5, respectively. The results of Experiment 4.

| Dataset | Model | $\frac{s}{\text{Image Size}}$ | $\frac{C_c(g)}{C_c(f_j)}$ | $\frac{C_s(g)}{C_s(f_j)}$ | Accuracy for $(N, T)$ | | |
|---|---|---|---|---|---|---|---|
| | | | | | $(2, 1)$ | $(3, 2)$ | $(4, 3)$ |
| MNIST | N(Iden. → Iden.) | 1 | 0 | 0 | 90.72 | 90.52 | 90.23 |
| | N(Iden. → 32) | 1 | 3.1e-5 | 5.1e-5 | 95.16 | 95.10 | 94.87 |
| | N(Iden. → 64) | 1 | 6.3e-5 | 9.9e-5 | 96.40 | 95.57 | 94.82 |
| | N(1 → Iden.) | 1.0e-1 | 3.1e-4 | 4.0e-6 | 90.53 | - | - |
| | N(2 → Iden.) | 2.1e-1 | 6.2e-4 | 8.1e-6 | 94.21 | - | - |
| Fashion-MNIST | N(Iden. → Iden.) | 1 | 0 | 0 | 81.00 | 80.45 | 80.13 |
| | N(Iden. → 32) | 1 | 3.1e-5 | 5.1e-5 | 83.32 | 82.45 | 82.04 |
| | N(Iden. → 64) | 1 | 6.3e-5 | 9.9e-5 | 84.00 | - | - |
| | N(Iden. → 128) | 1 | 1.2e-4 | 1.9e-4 | 83.02 | - | - |
| | N(2 → Iden.) | 2.1e-1 | 6.2e-4 | 8.1e-6 | 81.69 | - | - |
| | N(4 → Iden.) | 4.1e-1 | 1.2e-3 | 1.6e-5 | 83.35 | - | - |
| | N(8 → Iden.) | 8.3e-1 | 2.4e-3 | 3.2e-5 | 81.51 | - | - |
| | N(4 → 32) | 4.1e-1 | 1.3e-3 | 6.7e-5 | 83.13 | - | - |
| Cifar-10 | N(Iden. → Iden.) | 1 | 0 | 0 | 35.70 | 35.47 | 35.12 |
| | N(Iden. → 32) | 1 | 2.3e-5 | 3.9e-5 | 38.30 | - | - |
| | N(Iden. → 64) | 1 | 4.7e-5 | 7.6e-5 | 42.47 | - | - |
| | N(Iden. → 128) | 1 | 9.3e-5 | 1.5e-4 | 42.28 | - | - |
| | N(8 → Iden.) | 2.6e-1 | 6.7e-3 | 7.2e-5 | 50.30 | - | - |
| | N(16 → Iden.) | 5.2e-1 | 1.3e-2 | 1.4e-4 | 54.39 | - | - |
| | N(32 → Iden.) | 1.0 | 2.2e-2 | 2.9e-4 | 58.13 | - | - |
| | N(64 → Iden.) | 2.1 | 3.7e-2 | 5.7e-4 | 55.27 | - | - |
| | N(32 → 128) | 1.0 | 2.2e-2 | 4.3e-4 | 58.16 | 58.02 | 57.63 |

In this table, the difference in the cost of the client and each server (in terms of computational and storage complexity) is quite evident. Also, the communication load between the client and each server is low, which is in the order of the size of $\boldsymbol{X}$ (in the stage of sending query), and in the order of the size of $\boldsymbol{Y}$ (in the stage of receiving answer).

# E  RELATED WORKS

In this section, we present privacy concerns in ML and the techniques that can be used to preserve privacy.

**Data privacy concerns in ML:** Offloading ML tasks to the cloud servers raises the concern of maintaining the privacy of the used datasets, either *the training dataset* or *the user dataset*, such that the level of information leakage to the cloud servers is under control.

The information from the training dataset may leak either during *the training phase* or from *the trained model*. In *training phase privacy*, a data owner that offloads the task of training to some untrusted servers is concerned about the privacy of his sampled data. In *trained model privacy*, the concern is to prevent the trained model from exposing information about the training dataset. On the other hand, in *inference privacy*, a user wishes to employ some server(s) to run an already trained model on his dataset while preserving the privacy of his dataset against curiosity of servers. Note that in inference privacy, there is an additional concern that we keep the computational burden on the user/client-side low.

**Remark (Inference privacy vs. differential privacy settings):** It is worth noting that inference privacy, the focus of this paper, and trained model privacy, the problem that mostly deals with differential privacy techniques, are two different problems with different goals.

The trained model privacy problems have these steps: 1) in the training phase, an individual, say Alice, locally trains a model on his sensitive training dataset such that the information leakage of the training dataset from the trained model is negligible; 2) Alice releases the trained model to a client, say Bob, in the form of black-box or white-box; 3) in the inference phase, Bob locally uses this model to label his sensitive data. The goal is to limit the information leakage of Alice's training dataset to Bob from the trained model while maintaining the inference accuracy high (note that since the inference phase performs locally, no information is leaked from Bob's data). As you can see, no phase of training or inference is offloaded to

cloud servers to reduce the computational load of the parties. In other words, in the conventional settings that mostly deals with differential privacy, there is no opportunity to offload computational tasks to servers while preserving privacy, since these settings are not basically designed for this purpose. Instead, in inference privacy, we have such a purpose.

The inference privacy problem has these steps: 1) in the training phase, a client trains a model on his own or public training dataset using some servers; 2) in the inference phase, he labels his sensitive data with the aid of the servers. The goal is to limit the information leakage of the client data in the inference phase while maintaining the inference accuracy high and keeping computational costs at the client low. In the training phase, the system is trained/designed for such a goal.

**Privacy preserving techniques:** Various methods have been proposed to protect privacy in the literature, with the following three major categories:

**Randomization, Perturbation, and Adding Noise:** Those techniques can be used for the trained model, training phase, and inference privacy at the cost of sacrificing the accuracy of the results.

In (Fredrikson et al., 2015; Shokri et al., 2017), it is shown that parameters of a trained model can leak sensitive information about the training dataset. Various approaches based on the concept of differential privacy (Dwork, 2006; Dwork et al., 2006; Dwork & Roth, 2014) have been proposed to reduce this leakage. A randomized algorithm is differentially private if its output distributions for any two adjacent input datasets, i.e., two datasets that differ only in one element, are close enough. This technique has been applied to principal component analysis (PCA) (Chaudhuri et al., 2013; Dwork et al., 2014), support vector machines (SVM) (Rubinstein et al., 2012), linear and logistic regression (Chaudhuri & Monteleoni, 2009; Zhang et al., 2012), deep neural networks (DNNs) (Abadi et al., 2016; Papernot et al., 2017), and distributed and federated learning (Shokri & Shmatikov, 2015).

A line of works on the federated learning setting (Heikkilä et al., 2017; Bonawitz et al., 2017; Jayaraman et al., 2018; Imtiaz et al., 2019a;b; Sabater et al., 2020), exploiting the existence of a cluster of non-colluding data owners and servers, utilizes techniques from MPC for secure aggregation to increase their performance and privacy. In particular, each party has a sensitive dataset, and they want to do differentially private computations (e.g., training an ML model) jointly across all datasets. The noise of each party is obtained from the sum of an independent term and a correlated term. Their system is designed such that the correlated noises are completely eliminated in the output, and only the independent noises remain to produce a differentially private result. Although their method improves performance, still the remained noise sacrifices accuracy. The caveat is that the privacy-accuracy tradeoff in those approaches (Alvim et al., 2011) bounds the scale of randomization and thus limits their ability to preserve privacy. Note that the purpose of the above papers is not to reduce the computational burden on the users, no ML tasks are offloaded to cloud servers, and their scope falls into the trained model privacy problem.

InstaHide (Huang et al., 2020), using an instance-hiding scheme, releases an encrypted version of the training dataset to perform the training phase with privacy preserving.

The client can offload some computational tasks during the inference phase of a deep neural network by partitioning it between the client and the cloud (Li et al., 2017). Despite the fact that revealing transformed data prevents direct disclosure of the raw data, valuable information still leaks to the servers (Wang et al., 2018a); therefore, some privacy-protective actions (e.g., adding noise) are needed to be taken. Authors in (Osia et al., 2018; 2020) perturb the data by passing it through the local module trained to protect some sensitive attributes of the data while preserving the accuracy of learning as much as possible. As an alternative approach, (Mireshghallah et al., 2020; 2021), without retraining the entire network, train only the noise distribution to protect privacy. Inspired by Generative Adversarial Networks (GANs) (Goodfellow et al., 2020), several works (Oh et al., 2017; Raval et al., 2017; Huang et al., 2017; Wu et al., 2018; Leroux et al., 2018; Ren et al., 2018; Chen et al., 2018; Kim et al., 2019; Tripathy et al., 2019) attempt to generate queries such that a computationally bounded adversary cannot extract some sensitive information from them. The performance of those techniques relies on heavy computation on the client-side, which is not desirable.

*K-anonymity* (Samarati & Sweeney, 1998; Sweeney, 2002; Bayardo & Agrawal, 2005) is another privacy preserving framework, in which the data items, related to one individual cannot be distinguished from the data items of at least $K-1$ other individuals in the released data. It is known that $K$-anonymity framework would not guarantee a reasonable privacy, particularly for high-dimensional data (Aggarwal, 2005; Brickell & Shmatikov, 2008).

**Secure Multiparty Computation:** By exploiting a cluster of non-colluding servers, this approach protects data privacy in ML algorithms that can be represented or approximated with a particular class of functions (e.g., polynomials) in each

Table 4: Techniques and Problems

| Technique | Privacy Problem | Method |
|---|---|---|
| Randomization, Perturbation, and Adding Noise | Trained model | (Chaudhuri et al., 2013; Dwork et al., 2014; Rubinstein et al., 2012) (Chaudhuri & Monteleoni, 2009; Zhang et al., 2012; Abadi et al., 2016) (Papernot et al., 2017; Shokri & Shmatikov, 2015; Heikkilä et al., 2017) (Bonawitz et al., 2017; Jayaraman et al., 2018; Imtiaz et al., 2019a) (Imtiaz et al., 2019b; Sabater et al., 2020; Huang et al., 2020) |
| | Training phase | (Huang et al., 2020) |
| | Inference | (Li et al., 2017; Liu et al., 2017a; Wang et al., 2018a; Osia et al., 2018) (Osia et al., 2020; Mireshghallah et al., 2020; 2021; Oh et al., 2017) (Raval et al., 2017; Huang et al., 2017; Wu et al., 2018; Leroux et al., 2018) (Ren et al., 2018; Chen et al., 2018; Kim et al., 2019; Tripathy et al., 2019) |
| Multiparty Computation | Training phase | (Gascón et al., 2017; Chen et al., 2019; Mohassel & Rindal, 2018) (Wagh et al., 2019; Patra & Suresh, 2020; Byali et al., 2020; So et al., 2019) (Wagh et al., 2020; Koti et al., 2021; Mohassel & Zhang, 2017) |
| | Inference | (Gascón et al., 2017; Chen et al., 2019; Mohassel & Rindal, 2018) (Wagh et al., 2019; Patra & Suresh, 2020; Byali et al., 2020; So et al., 2019) (Wagh et al., 2020; Koti et al., 2021; Mohassel & Zhang, 2017) (Liu et al., 2017b; Juvekar et al., 2018; Riazi et al., 2018; 2019) |
| Homomorphic Encryption | Training phase | (Graepel et al., 2012; Wang et al., 2018b; Han et al., 2019) |
| | Inference | (Graepel et al., 2012; Gilad-Bachrach et al., 2016; Hesamifard et al., 2017) (Li et al., 2018; Han et al., 2019) |

round (Mohassel & Zhang, 2017; Gascón et al., 2017; Mohassel & Rindal, 2018; So et al., 2019; Chen et al., 2019; Wagh et al., 2019; Patra & Suresh, 2020; Byali et al., 2020; Wagh et al., 2020; Koti et al., 2021). In several attempts (Liu et al., 2017b; Juvekar et al., 2018; Riazi et al., 2018; 2019), MPC and cryptography tools have been leveraged to provide oblivious inference in which neither the servers learn the client data nor the client learns the servers' model. The shortcoming of MPC is that it costs the network a huge communication overhead. Moreover, the particular setups of those schemes (e.g., polynomial approximation and finite field arithmetic) make them challenging to apply to deep learning. In contrast, our approach is applicable to modern ML models.

**Homomorphic Encryption:** HE enables a cryptographically secure environment between the client and servers to perform ML algorithms (Graepel et al., 2012; Hesamifard et al., 2017; Li et al., 2018; Wang et al., 2018b). The public-key infrastructure in HE schemes imposes heavy computational overhead on the system. This disadvantage is directly reflected in the time needed to train a model or use a trained one, as reported in (Gilad-Bachrach et al., 2016; Han et al., 2019). Besides, those schemes are based on computational hardness assumption, not information theory.

**Other techniques:** Some approaches that do not fall into the above categories are described in the following:

*Trusted Execution Environment (TEE):* TEEs, e.g., ARM TrustZone (Alves, 2004) and Intel SGX (McKeen et al., 2013), provide an execution space where data can be processed in an isolated environment. By embedding TEEs on an untrusted server, ML algorithms can be deployed in a secure and private environment between the client and the server (Ohrimenko et al., 2016; Hunt et al., 2018; Narra et al., 2019; Hanzlik et al., 2021). Despite having more functionality and outperforming HE schemes, TEEs have low availability and performance compared to rich untrusted alternatives (e.g., GPUs), making them challenging to use (Tramer & Boneh, 2018).

*DataMix (Liu et al., 2020):* This paper suggests a privacy protection scheme inspired by mixup (Zhang et al., 2017). The client mixes $N$ images and sends $N$ queries to one server - each a distinct linear combination of those images - and then de-mixes the server outputs to predict the image labels. However, the scheme becomes prone to privacy violations if all the images are sensitive since the linear combinations are disclosed to the server. On the other hand, if some of those images are not sensitive and are used as noise to confuse the server, privacy will scale with communication; the more images we mix, the more queries we need to send to the server.

Table 4 categorizes the papers according to technique and problem.