# OpenReview forum: "Trained-MPC: A Private Inference by Training-Based Multiparty Computation"
_MLSys/2023/Workshop/RCLWN — MLSys-RCLWN 2023_

### Official Review · Reviewer_bV31 · 2023-04-20

**Rating:** 6
**Confidence:** 3

**Review:**

**Overall summary**: The paper proposes an approach to tackle the 'inference privacy' problem. This problem is motivated by resource-constrained devices, which need to send their data to the servers in order to perform some inference tasks. The authors present a privacy-preserving approach, motivated by multi-party computation and the addition of noise.  They test out their method on some well-known datasets as well as compare it wrt. a comparable previous work called ARDEN.

Remarks:

1. While reading the introduction section, I felt that some important concepts were briefly and vaguely mentioned, without going into important details. I believe this has debilitated the credibility of the contributions and the motivation for the system model depicted in Figure 1.

For example, in **(I) Randomization, Perturbation, and Adding Noise** subsection, it is unclear, yet to the reader, what the authors mean by *information leakage* -- there are various definitions of this concept in the literature (and also related references are missing..).

In **(II) Secure Multiparty Computation (MPC)** and in **(III) Homomorphic Encryption (HE)** subsections, it is unclear why they pose significant communication and computational overhead, (and therefore, should *not* be preferred by the resource-constrained edge devices..).

Although some of the above concepts are detailed in the appendix, I felt like the write-up of the introduction can be polished in order to better introduce the problem landscape that the proposed method is operating in.

2. The related work that the authors compare their method with, i.e. ARDEN, is not mentioned anywhere in the introduction -- it is not completely clear which type of technique it employs wrt. others in the literature. I also felt like the presentation and comparison with the ARDEN were a bit too late in the text as well as rather brief.

3. The overall presentation and write-up of the paper can be further improved. There are some ill-posed sentences such as:

> Thanks to the very strong noise with a standard deviation of 70 times the data,...

> Also, the threat model assumes any adversary in the servers has unlimited computational resources to extract information about
the client data. In other words, we preserve information theoretic privacy in this paper.

4. Some important references are missing, such as Shannon's mutual information below **Definition 1** and differential privacy in **Definition 2**.  References to the datasets (MNIST, Fashion-MNIST, CIFAR-10) are also missing.

5. While the authors explain the *operational* meaning of differential privacy just below **Definition 2**, a similar intuitive explanation for mutual information case is missing -- it is unclear to the reader what it effectively means to bound the mutual information term with $\epsilon$ as in **Definition 1**.

6. I liked the general idea of the Trained-MPC and the motivating example illustrated in Figure 1. The mathematical formulation at the end of the introduction section is quite clear.

7. Looking at Figure 3, we can see the effectiveness of the Trained-MPC method considering mutual information privacy. The authors try different variants of the approach, such as I-III, having different layer configurations of the client's neural network. It would have been nice to see a *quantitative comparison*  regarding the computational complexity across these variants. Also, it would have been convincing to see some experimental results regarding differential privacy in the main text, as the authors dedicate a space of one definition and one theorem for this variant of privacy.

8. Overall, I felt like although there are important and well-thought-of ideas/concepts presented in the paper, the current presentation style makes it hard to follow the discussion and understand how this approach is effectively different wrt. others in the literature. I would suggest that the authors try to polish some of the sections, in the main text, in order to make the paper accessible to the audience of this workshop.

---

### Meta-Review · Area_Chair_moGm · 2023-05-03

**Recommendation:** Accept
**Confidence:** 4

**Metareview:**

The paper proposes a technique for private inference, the general idea is interesting and relevant.
Several important points have been raised by the reviews to improve presentation and clarify the contributions with respect to prior work. please revise the final version accordingly.

I have an additional technical comment regarding formulation (1): Why does the leakage constraint only take into account the features, and not the labels? As training is done over the true labels, functions f1_, ..., fN may memorize the correlation between noisy features and true labels, essentially denoising?